# Influenza A Outbreaks in Two Professional Ice Hockey Teams during COVID-19 Epidemic

**DOI:** 10.3390/v14122730

**Published:** 2022-12-07

**Authors:** Niklas Lindblad, Timo Hänninen, Maarit Valtonen, Olli J. Heinonen, Matti Waris, Olli Ruuskanen

**Affiliations:** 1Unit for Health and Physical Activity, Paavo Nurmi Centre, University of Turku, 20520 Turku, Finland; 2Tampere Research Centre of Sports Medicine, UKK Institute for Health Promotion Research, 33500 Tampere, Finland; 3Finnish Institute of High Performance KIHU, 40700 Jyväskylä, Finland; 4Institute of Biomedicine, University of Turku, 20520 Turku, Finland; 5Department of Clinical Microbiology, Turku University Hospital, 20521 Turku, Finland; 6Department of Paediatrics and Adolescent Medicine, Turku University Hospital, 20521 Turku, Finland

**Keywords:** influenza, respiratory virus, ice hockey, exercise, athlete

## Abstract

Influenza A outbreaks occurred in two professional hockey teams just after two games they played against each other. Thirteen players and two staff members fell ill during 17–20 April 2022, while COVID-19 was prevalent. Altogether, seven players missed an important game due to influenza. The rapid diagnosis permitted effective pharmaceutical and nonpharmaceutical control of the outbreaks.

## 1. Introduction

Acute respiratory infection (ARI) is the most common illness in elite athletes. The aetiology and disease burden of ARIs in athletes is not well-studied and many questions remain. The real occurrence of ARI in athletes is not known because there are no high-quality long-term studies. Surprisingly, there are only few aetiological studies conducted in athletes with ARI, and the clinical manifestations are poorly documented. Furthermore, risks to health while exercising with ARI are not known and return-to-sports protocols are non-scientific [1,2,3]. The still ongoing COVID-19 pandemic has had an unforeseen impact on professional sport activities [3,4]. In the Finnish ice hockey leagues, the 2019–2020 season was interrupted, while during the 2020–2021 season, teams with SARS-CoV-2 infections were temporarily quarantined. In the 2021–2022 season, symptomatic players were tested for SARS-CoV-2 and restrictions concerned only players with a positive test result. Focusing on only one pathogen, other important infections may become neglected. We report influenza A outbreaks in two Finnish Elite League ice hockey teams who played against each other twice immediately before the outbreaks. There are no previous studies on non-COVID respiratory virus infections in ice hockey players.

## 2. Materials and Methods

The study population consists of two professional ice hockey teams playing in the Finnish Elite League. At the time of the study, Team I had 25 players and 10 staff members and Team II had 27 players and 10 staff members. During April 2022, the teams played 4 semi-final games against each other (Figure 1). For influenza diagnostics, rapid antigen detections were used: Actim Influenza A&B (Medix Biochemica, Espoo, Finland) in Team I and mariPOC Respi+ (ArcDia International LTD, Turku, Finland) in Team II. A multiplex PCR test for 16 respiratory viruses: influenza A and B viruses; respiratory syncytial A and B viruses; adenoviruses; enteroviruses; metapneumoviruses; parainfluenza virus types 1–4; bocaviruses; coronaviruses 229E, NL63, and OC43; and rhinoviruses (Allplex Respiratory Panels 1–4, Seegene, Seoul, South Korea) was used for community surveillance and to verify the type of influenza. SARS-CoV-2 was searched from all symptomatic cases using tests of local authorities.

According to the Finnish Advisory Board on Research Integrity, the study did not need an ethical review.

## 3. Results

On 17 April 2022, a player in Team I developed a fever, a sore throat, and a cough. He reported the symptoms to the team physician (TH) the following day. Influenza A was diagnosed on 19 April using a rapid antigen test. Treatment with oseltamivir was initiated. The fever lasted for three days. On 24 April, 6 days after the onset of influenza, he played a match. He considered his performance normal.

Three further players of confirmed febrile influenza were recorded on 17, 18, and 20 April, respectively. They were treated with oseltamivir. All the other team members were treated with oseltamivir as post-exposure prophylaxis. In addition, five players (antigen test negative) and one staff member (antigen test positive) developed symptoms of influenza on 19–22 April (Figure 1).

Six of the nine players played on 24 April, 4–7 days after the onset of influenza. Seven of nine players were not vaccinated against influenza.

Between 18 and 19 April, 2–3 days after the two games with Team I, two players in Team II reported to the team physician (NL) the development of a fever, a runny nose, and myalgia. Two other players and a staff member developed a fever and myalgia on 19 April. Players’ nasal mucus samples tested positive for influenza A using an antigen test on 20 April. Multiplex PCR test revealed an influenza A virus subtype H3N2 in three cases. Oseltamivir was administered as a treatment for the confirmed cases and as prophylaxis for the others. Strict mitigation measures were instituted, including enhanced hand hygiene, the use of personal water bottles, and the use of masks in the locker room.

All four players of Team II with influenza missed the game on 20 April but played on 23 April, 3–5 days after the onset of febrile influenza (Figure 1). Before returning to the sport, they had a medical check-up and a normal electrocardiogram. All players played four games during 23–28 April. Their performances were normal. The four players were vaccinated against influenza, i.e., they were vaccination failures. They were vaccinated 6 months earlier, which partly explains the failure.

## 4. Discussion

This is the first study reporting influenza outbreaks in professional ice hockey teams. It is strange because there are globally over one million registered ice hockey players, and about 80,000 of them may annually suffer influenza which may effectively spread within the team [5,6,7]. Importantly, influenza is an exceptional respiratory virus because it can be prevented by vaccination and specific antivirals, and it can be treated with specific antivirals. Furthermore, nonpharmaceutical measures, especially avoiding close contact with those ill, reduce transmission [8,9,10].

In this study, influenza A seasonal activity was atypical and as a cause of ARI, unexpected. Influenza activity was very low during the preceding weeks, whereas SARS-CoV-2 was highly prevalent. Furthermore, no other respiratory viruses were markedly circulating in the community (Figure 1). All ARI cases were SARS-CoV-2 negative. The rapid detection of influenza permitted effective oseltamivir treatment. The teams used antigen detection tests, which give results in 10–20 min. Antigen tests are easy to use and inexpensive but not always sufficiently sensitive (60–90%) in adults [8,9]. Positive results should be confirmed by PCR tests, which can identify aetiology in over 80% of symptomatic subjects with influenza [6,7]. The aetiology of ARI cannot be established on clinical grounds alone [6,7].

The rapid diagnosis of influenza permitted prompt mitigation procedures and the influenza outbreaks were effectively controlled, only lasting from 3–5 days. Ice hockey teams offer an ideal setting for the transmission of viral ARIs. Ice hockey is a close-contact indoor team sport in which the players shout and hug, often use common water bottles, and during breaks, sit closely side-by-side in the locker rooms, which may not be ventilated well enough. Furthermore, the teams travel frequently. Influenza transmitted to 15–20% of the players, mirroring the transmission rate in households [11]. With the influenza diagnosis, the medical team knew the incubation period (1–5 days), the generation time (3–4 days), and the duration of infectiousness (3–6 days). Team-to-team transmission was probable but remains unclear. Interestingly, in a recent study on COVID-19 in ice hockey teams, an asymptomatic carrier of SARS-CoV-2 infected 22 of 28 teammates. Altogether, 49 infections were recorded in five ice hockey teams that were in contact with each other [12].

Altogether, seven players missed an important game (three for bronze medals, four for gold medals) due to febrile influenza. An aetiologic diagnosis of ARI helped the decision as to when to return to sport. The mean time for returning to play in the six influenza cases was 4.3 days, which agrees with a recently reported recommendation of 2–4 days after ARI [2]. The health risks (sinusitis, pneumonia, myocarditis) of rapid return to sport after ARI have not been reported [1].

The limitations of our study must be acknowledged. The number of players was small, we did not study asymptomatic players, and finally, the mucus samples were not available for viral genomic analysis to confirm possible team-to-team transmission.

In conclusion, the rapid diagnosis of influenza permitted prompt pharmaceutical and nonpharmaceutical mitigation procedures in two ice hockey teams and the outbreaks were effectively controlled, only lasting from 3–5 days. Our observations support the view that virus diagnostics, especially for influenza, should be commonly used in athletes with ARI. It is worth remembering that the influenza virus is a cardiotoxic virus and may induce myocarditis and other systemic extrapulmonary complications [13,14]. The use of influenza vaccination in athletes should be stressed, although its efficacy may remain modest. In any case, vaccination may be associated with illness attenuation [15].

## Figures and Tables

**Figure 1 viruses-14-02730-f001:**
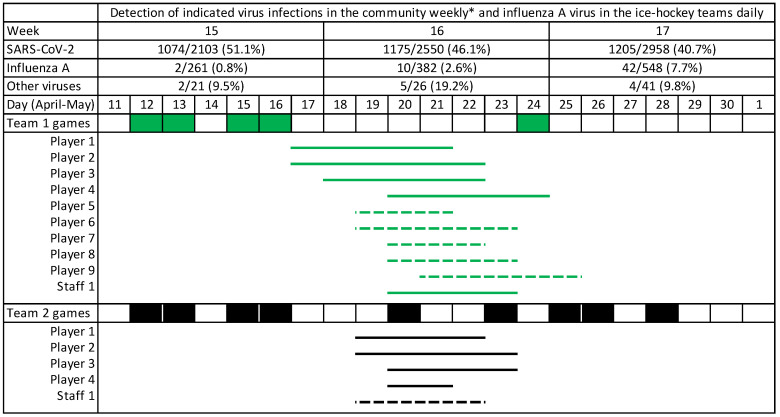
Timeline of influenza A virus outbreak in two ice hockey teams and background occurrence of respiratory viruses in the community as reported from the Hospital District of Southwest Finland. SARS-CoV-2 numbers result from community surveillance and diagnosis of hospital patients, numbers of influenza A and other viruses from hospital patients only. Teams played against each other on 12, 13, 15, and 16 April, thereafter against other teams. Lines indicate sick players with (solid line) or without (broken line) confirmed influenza A virus. Other viruses (n) included rhinovirus (4), human metapneumovirus (4), and seasonal coronaviruses 229E (2) and NL63 (1). * Positive specimens/tests performed.

## Data Availability

The data presented in this study are available on request from the corresponding author.

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
