# Peer review of "Influenza A Outbreaks in Two Professional Ice Hockey Teams during COVID-19 Epidemic"

_viruses, 2022, doi:10.3390/v14122730_

Round 1

Reviewer 1 Report

The manuscript of viruses-2067525 briefly reported influenza A outbreaks in 2 professional ice hockey teams. There are some suggestions here:

1. In the part of introduction, the effects of ARI on athletes are suggested to explain more. The authors can list some questions among "many questions remain".

2. 16 respiratory viruses (Allplex Respiratory Panels 1–4, Seegene, Seoul, South 39

Korea) are suggested to list.

3. In figure 1, SARS-CoV-2, Influenza A as well as other viruses should be explained the population investigated, since the numbers of people are different for all the three groups.

4. In line 65 Page 2, were "all four players" in team I or team II?

5. In line 68 Page 2, what does "The four players were vaccinated against influenza" mean?

Author Response

  1. In the part of introduction, the effects of ARI on athletes are suggested to explain more. The authors can list some questions among "many questions remain".

We have added remaining questions and one new reference [3] to the introduction. 

2. 16 respiratory viruses (Allplex Respiratory Panels 1–4, Seegene, Seoul, South 39 Korea) are suggested to list.

We have added the list of viruses to the methods section.

3. In figure 1, SARS-CoV-2, Influenza A as well as other viruses should be explained the population investigated, since the numbers of people are different for all the three groups.

We added an explanation into the Figure text.

4. In line 65 Page 2, were "all four players" in team I or team II?

Team II, added into the text.

5. In line 68 Page 2, what does "The four players were vaccinated against influenza" mean?

We modified the corresponding sentence as follows: The four players were vaccinated against influenza, i.e. they were vaccination failures. They were vaccinated 6 months earlier, which partly explains the failure.

Reviewer 2 Report

The manuscript, entitled “Influenza A outbreaks in 2 professional ice hockey teams during COVID-19 epidemic”, reported 13 players and 2 staff members from two teams have got acute respiratory infection during April 17-20, 2022. The text is OK, but the figure should be improved.

In the figure:

1. If the results of SARS-CoV-2, influenza A and other viruses were all from community? If any other pathogens rather than viruses had been identified using the multiplex PCR test, and the subtype of influenza A virus?

2. Clearly, the games and players belong to different teams. Please color the matches between these two teams, and the missed games.

3. Please add the two staff who developed symptoms of influenza.

4. Only one player from Team II was illness onset in April 19,  inconsistent with the text “Between April 18 and 19, 2-3 days after 2 games with Team I, 2 players in Team II …”.

Line 88: in the figure, influenza positive rate was 7.7% in Week 17. 7.7% is lower than 40.7% but is not “very low”. The authors could compare that with the influenza activity before COVID-19 pandemic.

Author Response

  1. If the results of SARS-CoV-2, influenza A and other viruses were all from community? If any other pathogens rather than viruses had been identified using the multiplex PCR test, and the subtype of influenza A virus?

No other pathogens were detected. 

2. Clearly, the games and players belong to different teams. Please color the matches between these two teams, and the missed games.

We added the information into the Figure text.

3. Please add the two staff who developed symptoms of influenza.

Only one staff member got infected, we added it to the Figure.

4. Only one player from Team II was illness onset in April 19,  inconsistent with the text “Between April 18 and 19, 2-3 days after 2 games with Team I, 2 players in Team II …

This mistake is corrected to the Figure, sorry about it.

Line 88: in the figure, influenza positive rate was 7.7% in Week 17. 7.7% is lower than 40.7% but is not “very low”. The authors could compare that with the influenza activity before COVID-19 pandemic.

Before week 15, no influenza cases were detected in the community. First cases were detected on week 15. We agree that 7.7% is not "very low", but that coincides with the outbreak.